# Differences in Emotional Preferences toward Urban Green Spaces among Various Cultural Groups in Macau and Their Influencing Factors

**Mengyao Wang, Yu Yan \*** , **Mingxuan Li** and **Long Zhou**

Faculty of Innovation and Design, City University of Macau, Macau SAR, China;
u22091120702@cityu.edu.mo (M.W.); u22091120588@cityu.edu.mo (M.L.); lzhou@cityu.edu.mo (L.Z.)
\* Correspondence: yuyan@cityu.edu.mo

**Abstract:** This study explores the diversity in emotional tendencies and needs toward urban green spaces (UGSs) among people from different cultural backgrounds in the wave of cultural integration. We utilized social media data as research tools, gathering a wide range of perspectives and voices. Utilizing geolocation data from 176 UGSs in Macau, we collected 139,162 social media comments to analyze the emotional perceptions of different cultural groups. Furthermore, we conducted regression analysis on the number of posts and emotional intensity values from four linguistic groups—Chinese, English, Southeast Asian languages, and Portuguese—in UGSs, correlating them with ten locally relevant landscape features. Our findings reveal diverse attitudes, emotional inclinations, and functional and design needs of different linguistic groups toward UGSs, as follows: (1) there were significant differences in emotional intensity and tweet counts across 176 UGSs; (2) Chinese and Portuguese speakers showed a more positive attitude toward plazas and natural ecological areas, whereas English- and Southeast-Asian-language speakers tended to favor recreational areas and suburban parks; (3) Chinese speakers exhibited a more positive emotional intensity toward sports facilities, while English speakers placed more emphasis on green space areas, architecture, sports infrastructure, and plant landscapes; (4) there was no specific landscape feature preference for Portuguese- and Southeast-Asian-language speakers. This research not only deepens our understanding of the emotional perceptions and preferences of UGSs among different cultural groups but also explores the association between these groups and various urban landscape features. This provides important theoretical and practical insights for future UGS planning, construction, and promoting multicultural coexistence for sustainable urban development.

**Keywords:** multicultural contexts; sentiment analysis; urban green space (UGS); social media data; landscape characterization

## 1. Introduction

Urban green spaces (UGSs) play a crucial role in sustainable development and human well-being [1,2]. The sustainable growth of cities is closely linked with human behavior and values [3,4]. As population growth, economic activities, and social and cultural interactions, as well as environmental and humanitarian impacts, become increasingly concentrated in urban settings [5], rural areas are becoming more domesticated, comprehensible, and industrialized. These transformations have led to a growing search for gardens and green spaces to compensate for the lack of natural wilderness in these areas [6]. As an essential component among a range of factors related to the quality of urban life, green spaces are gaining increasing attention [7].

Numerous studies have revealed the ways in which people interact with UGSs, dependent not only on the form, structure, and function of the spaces but also influenced by ethnic cultural identity [8,9]. Therefore, although many studies support the benefits of green spaces for mental health, the specific causal pathways are not yet fully clear [10],

and it remains unknown whether all types of green spaces have a positive impact on the emotions of those experiencing them. On the one hand, there may be features in UGSs that induce negative emotions [10], such as colors and symbols that may have different roles and symbolic meanings in different cultures [11,12]. On the other hand, the potential of landscape forms or key features in optimizing the expected benefits requires further exploration [13], especially in multicultural contexts in which identifying connections between culture and landscape features becomes more challenging. Therefore, understanding and recognizing emotional responses becomes a key task, revealing the impact of UGSs on human well-being through the dynamic responses of humans' emotions to their environment [14].

The core objective of this study was to analyze in depth the preferences and emotional responses of individuals to UGSs in a multicultural context. The research explores how people from different cultural backgrounds perceive and feel about the green spaces in the city and discusses which types of green spaces they are more likely to receive benefits. Additionally, this study aims to identify and compare the differences or similarities between different landscape features and the perceptions of users in a multicultural background. To achieve these research objectives, the study will focus on the following key questions:

RO1: What impact do UGSs have on people's emotions, and how can these impacts be quantitatively assessed?

RQ2: How do the emotional reactions people have to UGSs differ among groups from different cultural backgrounds?

RQ3: What specific landscape features influence the preferences for green spaces among groups from different cultural backgrounds?

## 2. Research Overview

### 2.1. Relationship between UGSs and Human Well-Being

Numerous studies have highlighted that UGSs are often overlooked in urban design [15–17]. Especially in densely populated cities, urban greenery, squeezed among the towering city structures, serves as one of the few relatively isolated yet vital natural habitats [18,19]. These spaces meet various physiological and psychological needs of diverse populations [20], providing an array of ecosystem services like improving air quality [21] and regulating urban microclimates [22]. They are not only carriers of historical memory but also treasuries of natural and cultural wealth [23]. Moreover, as a social construct, the sense of place evokes different reactions among heterogeneous groups—for some, a place might be romantic, while for others, oppressive; chaotic for some, peaceful for others; terrifying for some, safe for others [24,25]. UGSs, with their diverse characteristics, play a vital role in promoting mental relaxation, enhancing work productivity, and improving life quality for various needs [26]. Therefore, they are indispensable in supporting biodiversity and human well-being.

However, in the planning of UGSs, research often focuses on the biodiversity, environmental quality, and landscape features and how these factors influence visual quality and aesthetic appeal [27–31]. In contrast, people's knowledge, perceptions, and needs are less considered [32]. Most studies do not discuss the relationship between urban greenery and human well-being as a concept encompassing physical, psychological, and social aspects [33]. A fair bit of research has been conducted linking sociodemographic characteristics to perceptions of UGS in urban planning and landscape architecture. Refs. [34–37], research on populations from different cultural backgrounds is relatively scarce. These studies rarely delve into the views of multicultural groups on UGSs and link their emotional perceptions to specific landscape features. As UGSs significantly improve the living environment and promote human health, thereby positively impacting urban sustainable development [38,39], a more comprehensive and in-depth study of urban greenery is urgently needed.

## 2.2. Differences in the Experience of UGSs in a Multicultural Context

Urban planning faces the challenge of designing for a culturally heterogenous society. The current global trend of migration and integration has led to a significant proportion of immigrants and their descendants in today's urban populations [40]. Worldwide, the mix of people from various backgrounds and ethnicities relocating from one place to another has resulted in the diversity of social cultures and languages we see today [41]. The field of environmental psychology has long been devoted to studying the relationship between cultural and ethnic backgrounds and the perception of place [32]. Kahn posited that each country has specific cultures that are difficult to change significantly [42]. People's behaviors in environments are a function of past experiences and memories, values and beliefs, and local culture and history [34]. As early as 1992, Mesquita and Frijda concluded that emotions are both universal and culturally specific [43], suggesting that understanding emotional processes can help assess biological or cultural determinants. In 2003, they further found significant differences in emotional experiences across cultures [44], and in recent years, the cultural specificity in emotional expression has become a hotly debated topic [45,46].

The concept of happiness, linked to many personal aspirations, varies according to cultural and historical backgrounds, leading people from different cultural backgrounds to view and experience public spaces differently [47]. They offer diverse perspectives on the aesthetics of UGSs and their facilities and features based on their feelings [30]. The objective features of UGSs include basic amenities expected by tourists and residents [48,49], aesthetic and natural elements like plants and water features [50], and life features like animals [51]. People from different groups may have preferences for natural versus designed landscapes [52]. Facilities and amenities, as key features that enable various human activities [53], also influence emotional perception. The presence of facilities that meet recreational needs in UGSs typically generates positive emotions, while the lack thereof can be negative. Natural landscapes interact with human senses such as sight and hearing, and human–animal interactions can be viewed positively or negatively by urban residents [54]. For instance, unmanaged or quiet green spaces might serve as a "refuge" for stressed urban dwellers seeking tranquility, but they can increase anxiety in those who feel insecure, negatively impacting their emotions [47,55]. Thus, different social and cultural groups have diverse emotional perceptions of UGSs, and their preferences for different features, aesthetics, and recreational opportunities in various types of urban greenery can vary [35].

Despite extensive empirical research on the relationship between cultural specificity and emotional perception, there has been a lack of focus on the cognitive mechanisms involved in emotional expression and perception [56]. While the term "culture" is often included in the context of entire societies, it can also be studied among a range of different human groups [57]. This study aims to explore the relationship between emotional perception in a multicultural context and the features of UGSs to analyze how different cultural groups might use these spaces to alleviate negative emotions [58].

## 2.3. Perceptual Extraction of UGS Based on Sentiment Analysis

The types of emotions people feel are expressed through tweet texts, and the lexical meanings have a certain intensity of emotional connotations. Currently, relevant studies categorize emotions as positive, neutral, or negative [59,60]. To study emotional variations in different types of UGSs, regular surveys are necessary throughout the study process, which can be time-consuming for researchers and participants alike. To overcome these challenges of traditional methods, sentiment analysis technology, a noninvasive approach, offers a way to explore emotions expressed in user-generated content and can be applied on a large scale with ease [61,62]. These data are unstructured and includes text information reflecting the real emotions of urban residents (such as happiness, sadness, fear, disgust, anger, and surprise) and perceptions [63].

Subjectivity detection stands as a pivotal component of sentiment analysis, tasked with eliminating dispassionate "facts" or "neutral" comments. Yet sentiment classifiers are often overly tuned to fit sentences lacking in personal viewpoint into a constrained set of predefined categories [64], which can result in a lack of precision in emotion recognition. Consequently, enhancing the accuracy of sentiment analysis or integrating it more effectively with other methodologies to fulfill research aims persists as an area ripe for further investigation.

Sentiment analysis is the computational study of people's opinions, attitudes, and emotions toward entities [65]. Most sentiment analysis is conducted by collecting data from social media platforms [66], using natural language processing and text mining to extract information from the positive and negative words in the text, their context, and the linguistic structure of the text [67]. Current sentiment analysis technology has become mature and represents a growing trend [62]. However, most existing sentiment analysis focuses predominantly on the English language [68], with less attention to bilingual sentiment analysis [69], and even less to the analysis and attention of multiple languages. The advent and rapid growth of the Internet, social media, and other online forums have led to the continuous and rapid generation of text data, increasing the demand for sentiment analysis of data in different languages [70]. Therefore, cross-language sentiment analysis is an effective method to deal with diverse comments. By using sentiment analysis to identify and understand the differences in emotional tendencies in different languages, distinguishing the emotional needs of people from different cultural backgrounds, it is possible to analyze the correlation between positive emotions and landscape features, and determine how to enhance satisfaction with UGSs.

*2.4. Classification of Landscape Features in UGSs*

This research undertook a comprehensive analysis of UGSs in Macau, which showed that urban green spaces have positive effects on ecosystems and human health [71]. The study acknowledges the uneven distribution of different types of UGSs across the cityscape, leading to varying access and utilization by different user groups [72]. Given the increasing racial diversity in urban societies, the Environmental Justice movement emphasizes the importance of equitable access to natural spaces across different ethnic groups for sustainable urban development [73]. This is particularly pertinent in culturally diverse cities, where landscape preferences among multilingual communities show significant cultural differences. The ethnicity of these groups influences their usage preferences for UGSs, especially in terms of landscape features and activity needs [74,75].

Physically, the size of UGSs significantly impacts visitation frequency [76], and the perception of architectural environments is linked to changes in both the physical and mental health benefits gained from visiting these spaces [17]. Additionally, vegetation density and water bodies, as key natural elements, vary in their positive perception among users [77]. Studies have shown that vegetation density can create differentiated shaded areas in urban spaces, impacting the activities of visitors to these green spaces [78].

UGSs serve as public venues for physical activities, making open spaces in these areas vital for interaction and sports activities [79]. Diverse infrastructure types and comfortable pathways affect the types of activities that can be conducted, thus influencing the positivity of visits [80]. Furthermore, aesthetically pleasing plant landscapes and facilities like playgrounds can extend the duration of stay and are highly attractive to green space users [76]. UGSs also offer opportunities for interaction with nature. The variety of plant and animal species enriches the activities within these spaces and affects their attractiveness to visitors [51].

## 3. Materials and Methods

*3.1. Research Framework*

This research is segmented into three distinct phases, as illustrated in Figure 1. The initial phase is dedicated to preparatory tasks, which include data collection, literature

review, and onsite investigations. Following this, the research scope was refined on the basis of the preparatory work. For 176 UGSs, social media data were harvested from Twitter using geographic location information. These data were then categorized by language and subjected to sentiment analysis to determine the volume and emotional tone of tweets in different languages. Concurrently, landscape feature types, encompassing both physical and activity landscapes, were identified on the basis of literature studies. Subsequently, a correlation analysis was conducted between these landscape features and the quantity and sentiment of tweets in various languages. This analysis was complemented by the use of a geographic information system (GIS) to create visual spatial distribution maps. The ability to better visually represent the variability in the spatial distribution of the number of tweets and sentiments of different green space types on a map facilitated the analysis of the findings. The study culminated in an analysis of the results, leading to the formulation of conclusions.

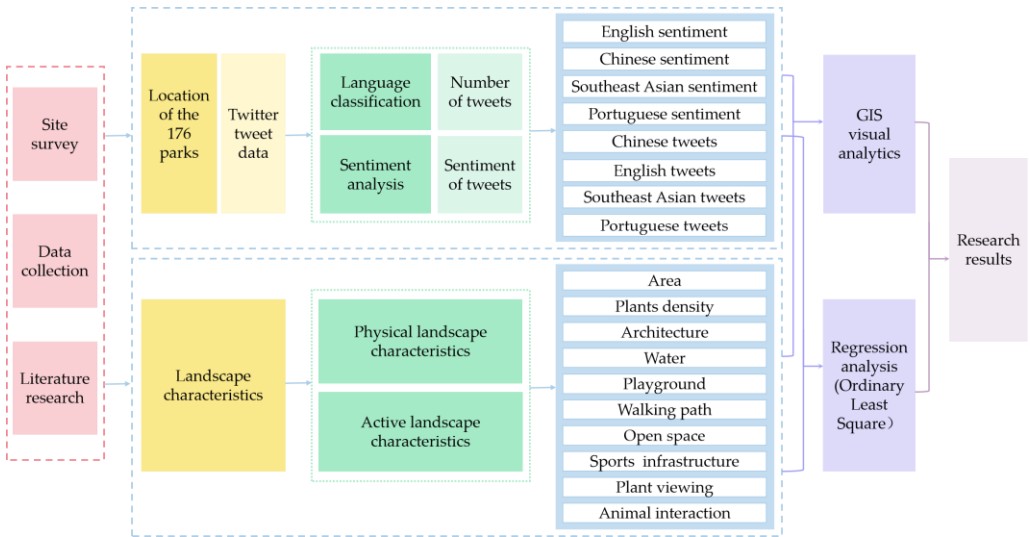

**Figure 1.** Research framework.

### 3.2. Research Area

This study focuses on the Macau Special Administrative Region of China as its primary research subject, as depicted in Figure 2. Macau, located at the confluence of mainland China and the South China Sea, adjacent to Guangdong province, presents a unique case study for several reasons. Firstly, Macau's rich historical tapestry has cultivated a multifaceted cultural milieu, blending Eastern Chinese traditions with influences from Portuguese and other Western cultures [81,82]. Officially using both Chinese and Portuguese languages, Macau offers a fertile ground for examining cross-linguistic emotional tendencies in green spaces among speakers of Chinese, English, and Portuguese. Secondly, its distinctive urban landscape, burgeoning tourism and entertainment industries, and the confluence of diverse cultural histories have made Macau a convergence point for migrant labor, tourists, and residents. This amalgamation enhances the relevance of Macau as a site for cultural and linguistic research within UGSs [83–85]. Lastly, as a typical high-density urban area, Macau's small geographical footprint and scarce natural resources limit its capacity for large recreational facilities [86], intensifying the demand for other leisure resources [87]. In such a context, the role of UGSs, vital for emotional regulation among urban residents, assumes even greater significance. In summary, Macau's selection as the research site for studying emotional inclinations in UGSs amidst a cross-cultural backdrop is not only typical of a diverse urban setting but also carries profound implications for research in this field.

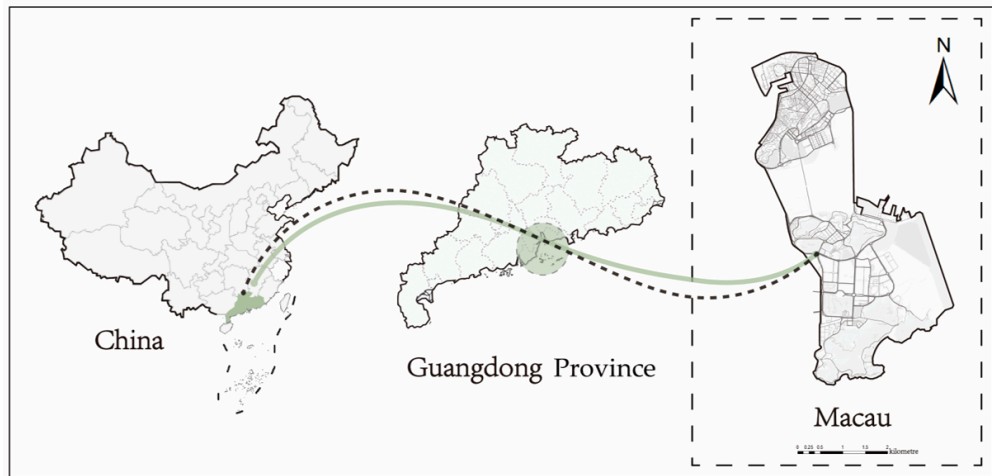

**Figure 2.** Location of the study area.

### 3.3. Park Type Classification

In this study, we initially selected categories of recreational facilities from the Macau Nature website, focusing on explicitly planned UGSs such as parks, gardens, country parks, rest areas, and natural ecological zones [88]. Through meticulous field surveys and extensive research, including data collection from the Macau Municipal Affairs Bureau website and other relevant sources [89], we expanded our selection to include viewing platforms and public squares as additional categories of UGSs for analysis. In total, 176 UGSs were chosen, encompassing 15.6% of the total area, as illustrated in Figure 3. This comprehensive selection of varied green spaces offers a robust foundation for our analysis of UGS utilization and its impact on diverse communities within Macau.

### 3.4. Data Collection

In June 2023, our study utilized data obtained from the Macau Land and Urban Planning Bureau's website, encompassing land use classification and vector data for 176 UGSs in Macau [90]. To gather social media data, we leveraged the geographical coordinates defining each green space, focusing on their latitude and longitude. Using these geographic locations as filters, we sourced tweets from Twitter's Advanced Search page (https://twitter.com/search-advanced, accessed on 7 December 2023). The data collection spanned from 17 September 2010 to 8 May 2023, encompassing posts from multiple Twitter users. This extensive process resulted in the acquisition of 139,162 geotagged tweets, providing a rich dataset for our analysis of public engagement with UGSs in Macau.

### 3.5. Statistical Analysis Method

3.5.1. Language Segmentation and Sentiment Analysis

In this study, the Twitter data were meticulously processed for language identification and sentiment analysis. Initially, we utilized "langdetect" (https://github.com/Mimino6 66/langdetect, accessed on 11 January 2024) for language recognition in the tweet text. This sophisticated tool, capable of identifying over 55 languages, operates on statistical techniques to accurately discern the language of a given text [91]. Each tweet was first preprocessed to prepare the data, following which the "langdetect" library was imported into our Python code. This library functionally classifies the language of each tweet by returning a language code string upon inputting the tweet text. The distribution of languages within our dataset was then meticulously analyzed, quantifying the count and proportion of tweets in various languages relevant to our study. We identified 11,314 tweets in Chinese, 75,010 in English, 5023 in Portuguese, 23,086 in Southeast Asian languages, and 24,729 in other languages.

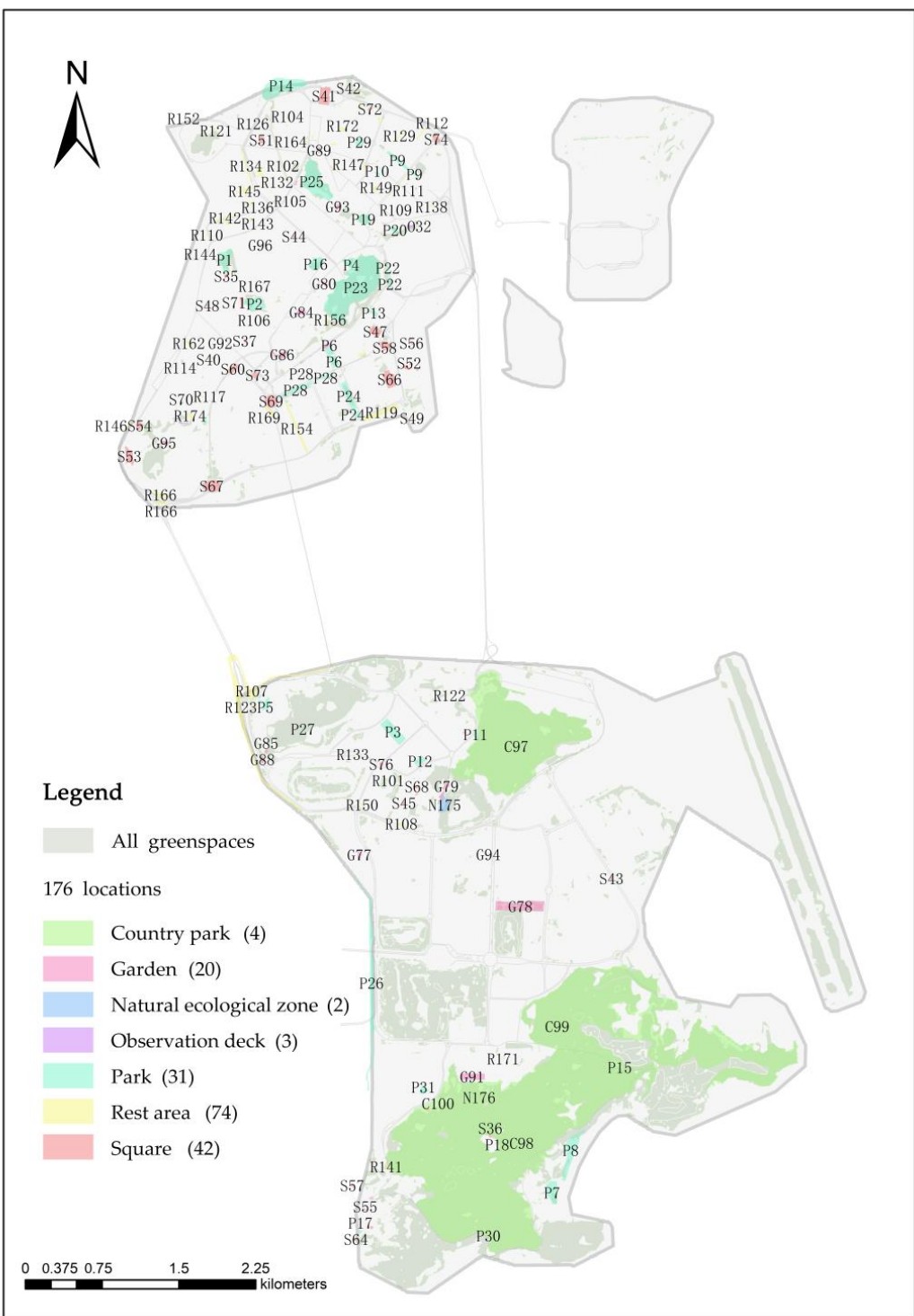

**Figure 3.** Public greenspace locations.

The sentiment analysis of these tweets was executed in two distinct steps, as follows: firstly, for English tweets and, secondly, for Chinese-, Portuguese-, and Southeast-Asian-language tweets. For English tweets, we employed the Natural Language Toolkit for sentiment analysis, which involved preprocessing steps like removing special characters, punctuation, and numbers, followed by text cleaning and standardization. The sentiment module of Python's Natural Language Toolkit library then assessed the emotional tone of the text, outputting a compound sentiment score. The detailed steps for evaluating the sentiment tone and awarding sentiment scores using the toolkit library are the following: firstly, the text is

segmented into words (or "tokens"); secondly, for each word, the word's sentiment score is looked up in its built-in sentiment lexicon, which contains about 7500 predefined words, each of which is assigned a score between −4 (very negative) and 4 (very positive); some heuristic rules are also considered, such as negatives (e.g., "not" or "never") changing the sentiment score of subsequent words, and enhancements (e.g., "not" or "never") changing the sentiment score of subsequent words (e.g., "very positive"); some heuristic rules are also considered, such as negative words (e.g., "not" or "never") change the sentiment score of the word that follows it, strong words (e.g., "very" or "extremely") increase the sentiment score of the word that follows it, and weak words (e.g., "kind of") decrease the sentiment score of the word that follows it. In the final step, the sentiment scores of all the words are added together to obtain the total sentiment score of the whole text, which can be used to determine the overall sentiment tendency of the text. A compound score above 0.05 was classified as positive, below −0.05 as negative, and scores within this range as neutral.

The second step involved using Google's Cloud Natural Language API for sentiment analysis of texts in other languages, undergoing similar preprocessing steps. The API was accessed using the "requests" library to obtain the probability of sentiment orientation (score) of each text. Scores above 0 were categorized as positive, below 0 as negative, and a score of 0 indicated neutral sentiment. This comprehensive and nuanced approach enabled a thorough sentiment analysis across different languages, vital for understanding the diverse emotional responses within our dataset.

### 3.5.2. Visual Analysis of Tweet Volume and Sentiment Intensity

In this phase of the study, a meticulous language categorization and sentiment intensity analysis were conducted on all tweets. Each city green space was then assigned an average sentiment intensity score, differentiated by language. To visually represent and analyze the volume and sentiment intensity of tweets for each UGS, this study employed ArcGIS 10.4 for the spatial analysis. Vector polygons representing the 176 UGSs were generated and attributed with field properties. In the process of symbolization, we adopted a quantity grading approach, employing the natural breaks (Jenks) method for color coding. This facilitated the creation of visual representations that intuitively depicted the distribution and sentiment trends associated with each green space across the urban landscape of Macau (Figure 4). This visual methodology not only enhanced the clarity of our findings but also provided an accessible means for interpreting the complex interplay between UGSs and the public's sentiment as expressed through social media.

### 3.6. Landscape Character and Ordinary Least Square (OLS) Regression Analysis

On the basis of a literature review, our study selected 10 landscape features for analysis, comprising four physical landscape features (area, architecture, plant density, and water) and six activity landscape features (playgrounds, walkways, sports facilities, open spaces, plant viewing, and animal interaction). These features were statistically analyzed to assess the quality of the 176 UGSs. The area of each green space was measured using ArcGIS 10.4 and categorized into 1–7 grades, the larger the value, the larger the area. Plant density was evaluated using 2023 aerial imagery of Macau and supplemented with field surveys, categorized into low, medium, and high densities. Other features like buildings, water, and playgrounds were also statistically assessed for their presence within the green spaces (Figure 5).

The study then employed OLS regression analysis to explore the relationships between the total number of tweets, the number of tweets in different languages, their sentiment intensities, and the 10 landscape features. This approach enabled a thorough exploration of how various UGS attributes correlate with the perceptions and emotional responses of diverse linguistic and cultural groups.

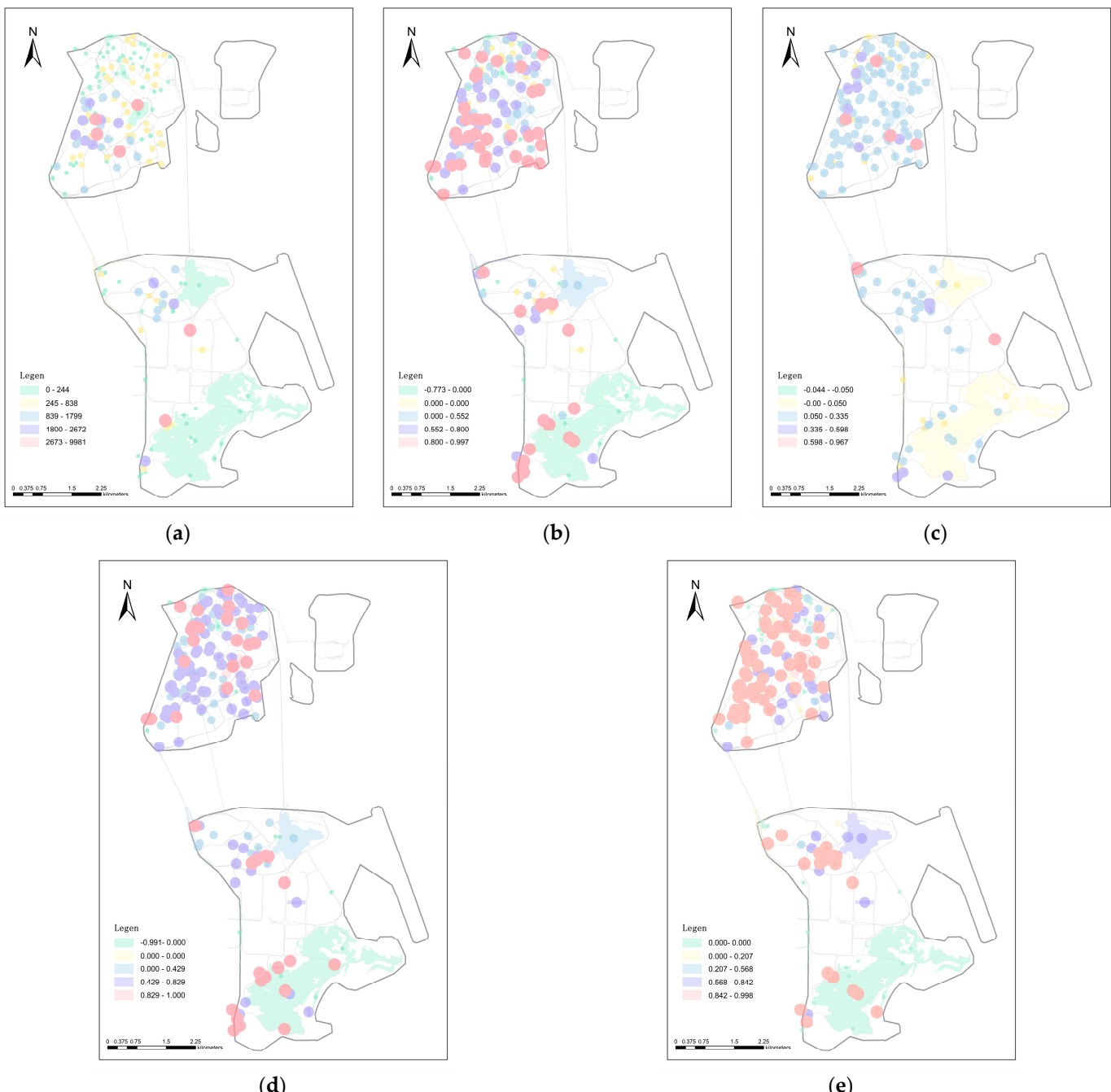

**Figure 4.** (**a**) Number of tweets; (**b**) Chinese tweet sentiment; (**c**) English tweet sentiment; (**d**) Southeast-Asian-language tweet sentiment; (**e**) Portuguese tweet sentiment.

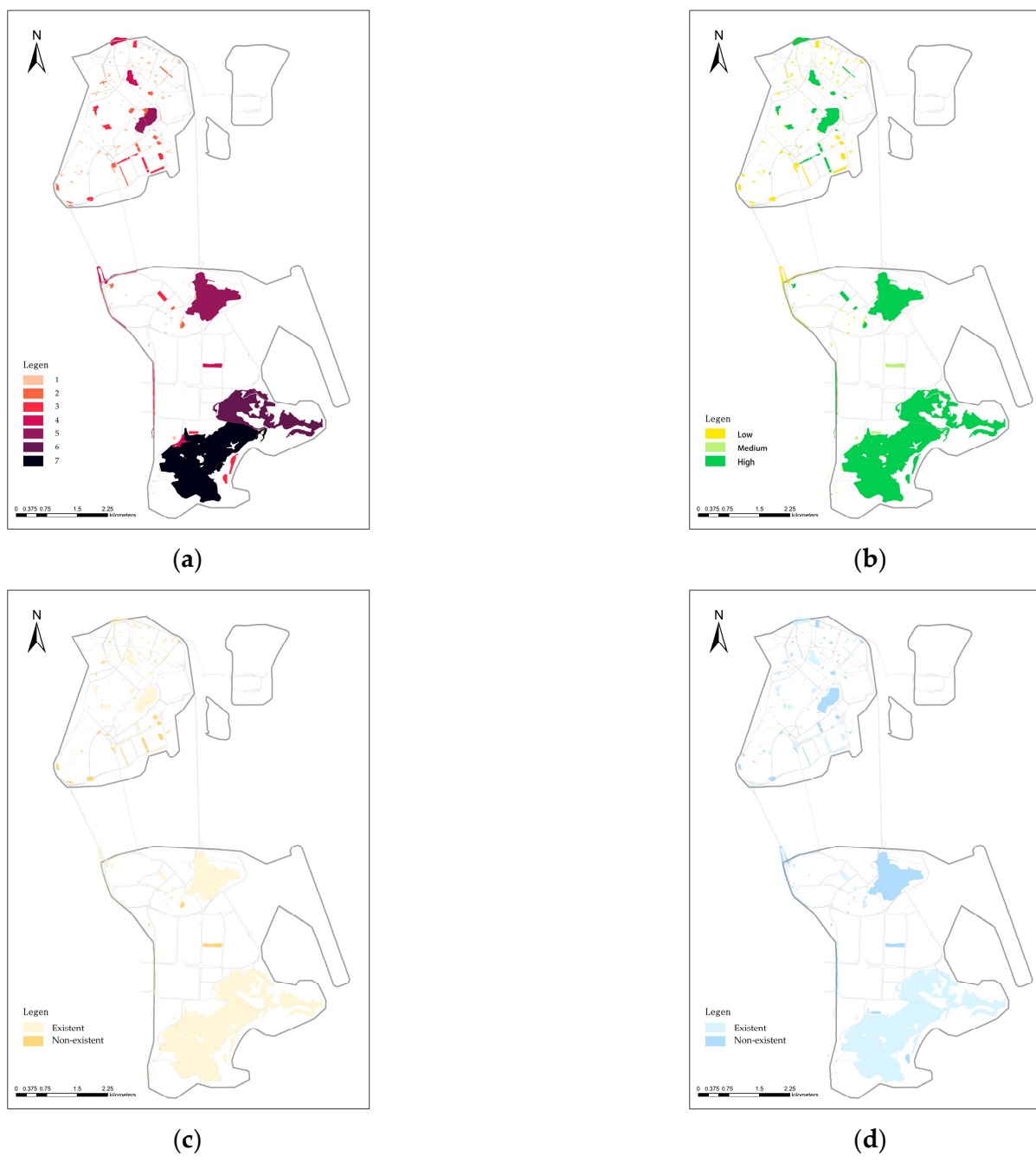

**Figure 5.** *Cont.*

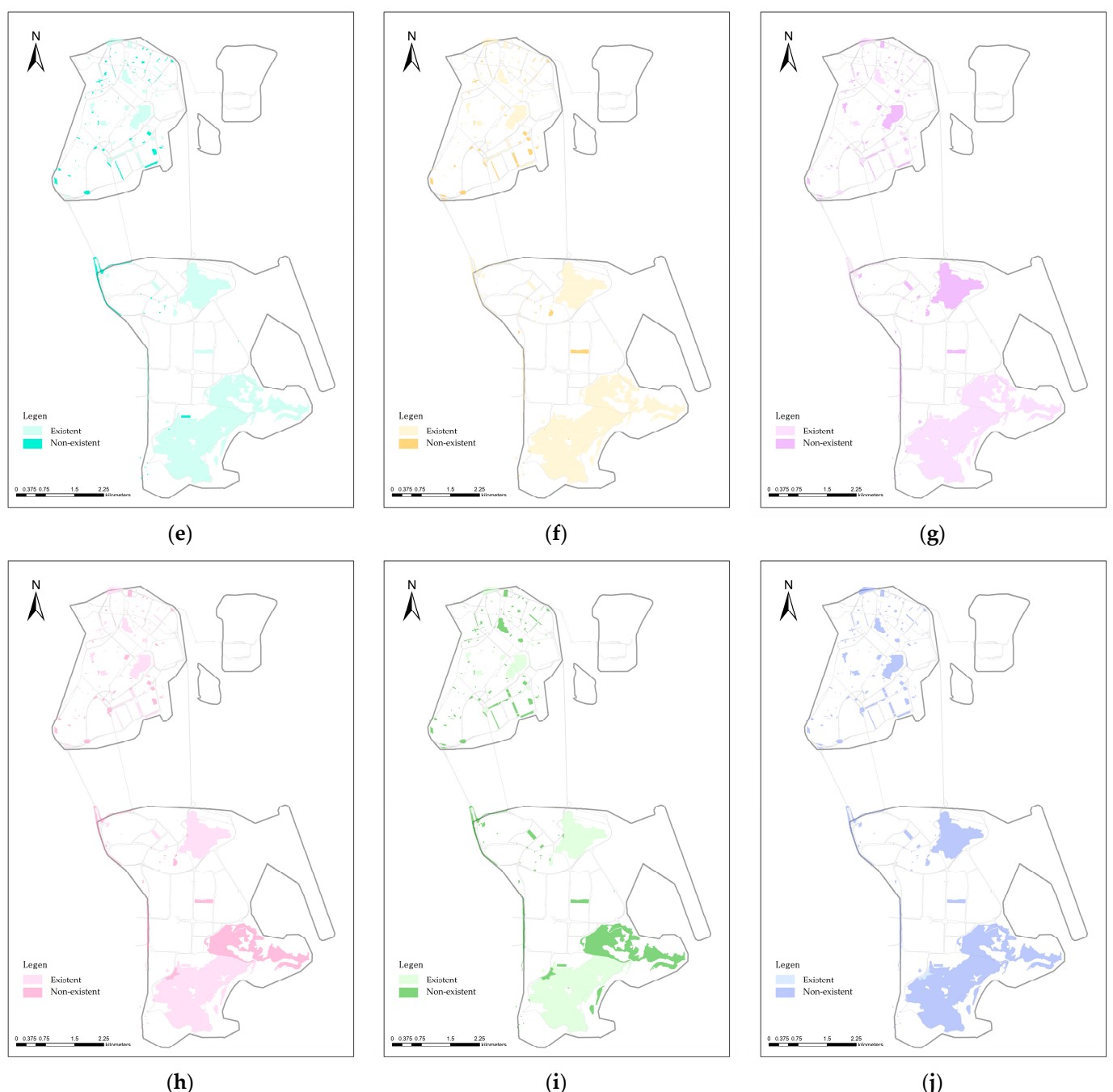

**Figure 5.** (**a**) Area; (**b**) plants density; (**c**) architecture; (**d**) water; (**e**) playground; (**f**) walking path; (**g**) open space; (**h**) sports infrastructure; (**i**) plant viewing; (**j**) animal interaction.

## 4. Results

### 4.1. Overall Sentiment Analysis of Different Types of UGSs

The analysis of the 176 UGSs in Macau (Figures 6 and 7) revealed significant insights into the volume of tweets and the average sentiment orientations across different languages. These data indicate that squares garnered the highest number of tweets, totaling 43,003. Parks followed closely with 30,595 tweets. While categories like nature reserves, viewpoints, and natural wetlands generated fewer than 10,000 tweets each, gardens and recreation areas exceeded this mark, with 16,114 and 23,913 tweets, respectively.

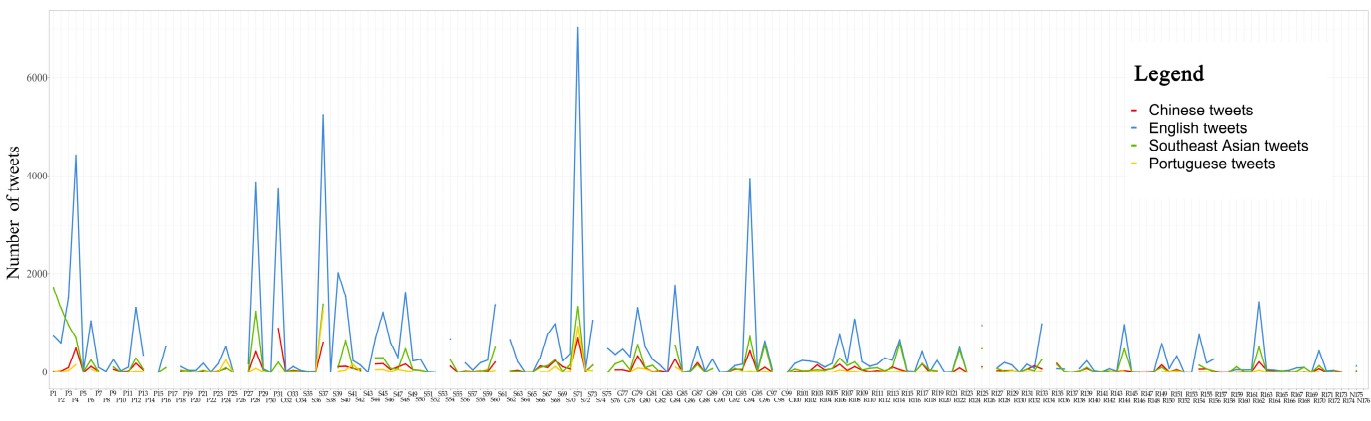

**Figure 6.** Number of tweets in four languages per green space.

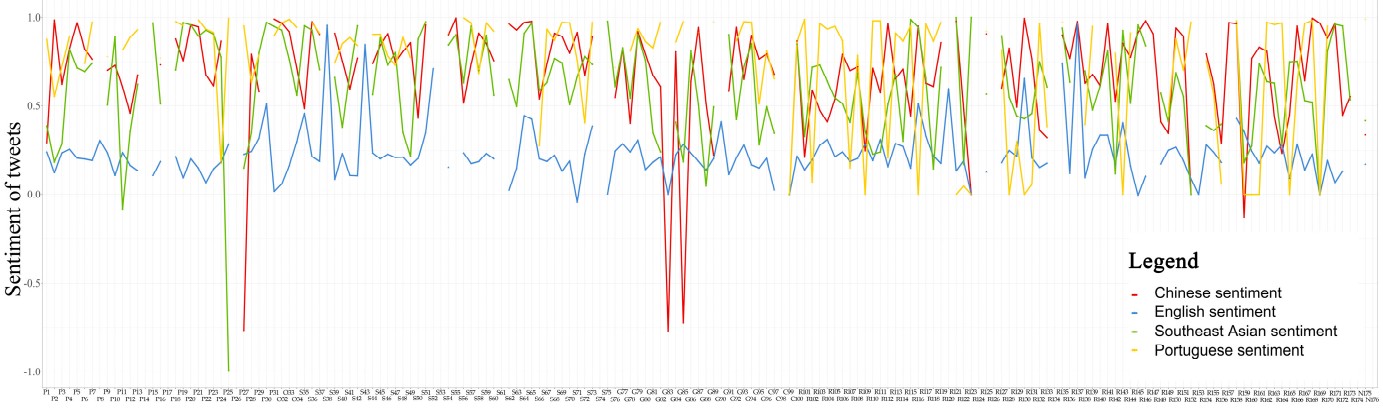

**Figure 7.** Emotion intensity values for four languages in each green space.

In terms of positive sentiment, viewpoints stood out with an average sentiment intensity of 0.429. Gardens and nature reserves followed with sentiment intensities of 0.392 and 0.364, respectively. Recreation areas, squares, and parks also displayed positive sentiment orientations, with values of 0.349, 0.325, and 0.306, respectively. The natural ecological areas registered the lowest sentiment intensity at 0.234.

These findings suggest that most types of UGSs provide positive emotional value to their users. However, because of the diversity in landscape features inherent in different types of green spaces, there are slight variations in sentiment intensities. Interestingly, it was observed that the green spaces with the most positive sentiment orientation were not necessarily those with the highest number of tweets. This highlights the complex relationship between user engagement, as reflected in social media activity, and the emotional value derived from different UGSs.

### 4.2. Sentiment Analysis of Tweets in Different Languages for Different Types of UGSs

The comparative analysis of tweets in Chinese, English, Portuguese, and Southeast Asian languages revealed distinct patterns in both the volume of tweets and their sentiment values. Specifically, Chinese tweets numbered 11,314 with an average sentiment value of 0.780, indicating a highly positive emotional orientation. English tweets, the most numerous at 75,010, displayed a relatively lower sentiment value of 0.175. Portuguese tweets, although fewer in number at 5023, registered a robust sentiment value of 0.804, suggesting a highly positive sentiment. Southeast-Asian-language tweets, totaling 23,086,

had a sentiment value of 0.537, which was moderately positive and higher than that of English tweets.

These data indicate that, in terms of positive sentiment orientation, Portuguese tweets exhibited the highest level of positive human emotional inclination, closely followed by Chinese tweets. Southeast-Asian-language tweets also showed a relatively positive sentiment, surpassing that of English tweets. This hierarchy of sentiment values underscores the varying degrees of emotional positivity expressed in different languages within the context of UGSs in Macau, reflecting a nuanced cultural landscape in terms of how different linguistic communities interact with and perceive these spaces.

The statistical analysis (Figures 6 and 7) demonstrates notable trends in tweet volume and sentiment intensity across different languages. English tweets, notably the highest in volume, predominantly focused on location S71. In contrast, the most frequent Chinese tweets centered around P31, while the highest number of Southeast-Asian-language tweets were concentrated in P1, and Portuguese tweets were most abundant at S37.

In terms of sentiment orientation, Chinese tweets exhibited a notably negative sentiment at locations P27, G83, G85, and R159. English tweets showed a similar negative sentiment trend at P11 and S71. Southeast-Asian-language tweets reflected negativity at P11 and P25. Portuguese tweets, however, maintained a consistently positive sentiment across all locations.

When considering the type of green spaces, tweets in Chinese, English, and Southeast Asian languages were most frequently posted about squares, whereas Portuguese tweets were predominantly about parks. The most positive sentiment in Chinese tweets was observed in observation deck, while Portuguese tweets showed a positive inclination toward country park. English tweets expressed the most positivity in rest areas, and Southeast-Asian-language tweets were most positive about natural ecological zones (Figures 8 and 9). This diversity in language-specific preferences and emotional responses underscores the cultural nuances influencing how different linguistic communities engage with and perceive UGSs.

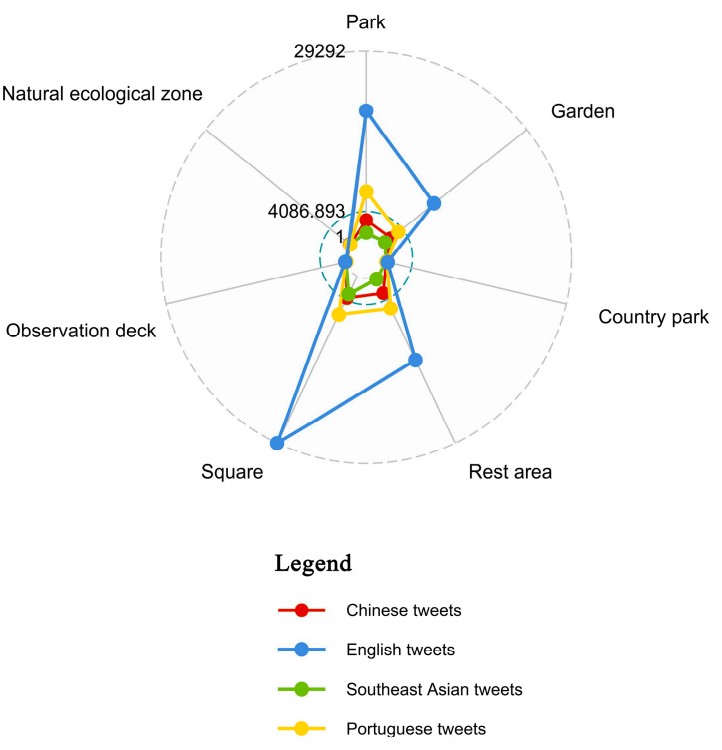

**Figure 8.** Number of tweets in four languages per type of greenspace.

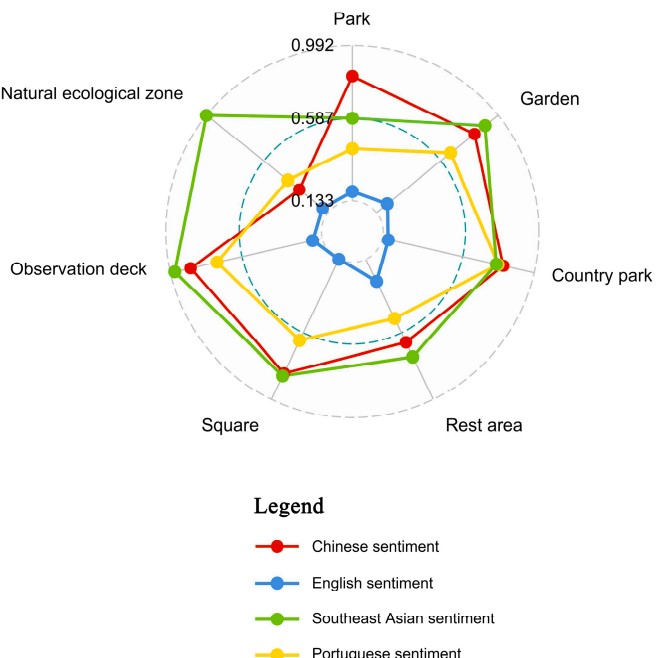

**Figure 9.** Emotional intensity values for four languages in each type of green space.

*4.3. Correlation Analysis of the Number of Tweets and Affective Tendencies with Landscape Features*

Our analysis of tweet volumes and sentiment across different languages, in relation to ten selected landscape features, reveals significant heterogeneity in landscape preferences among people from diverse cultural backgrounds. As demonstrated in Tables 1 and 2, the presence of water features showed the most substantial impact on tweet volume across all four languages, exerting a statistically significant negative influence. This pattern might be attributed to the generally small size of parks in Macau, where water features lack substantial aesthetic appeal, thereby diminishing visitors' inclination to post about them.

**Table 1.** Results of the *correlation* analysis of the number of tweets and landscape features.

| Landscape Characteristics | | Number of Tweets | | | | |
| --- | --- | --- | --- | --- | --- | --- |
| | | All Tweets | Chinese Tweets | English Tweets | Southeast Asian Tweets | Portuguese Tweets |
| Physical landscape characteristics | Area | −0.068 | −0.047 | −0.060 | 0.007 | −0.127 |
| | Plants density | 0.084 | 0.089 | 0.067 | 0.142 | 0.379 *** |
| | Architecture | 0.102 | 0.051 | 0.125 | 0.175 | 0.039 |
| | Water | −0.476 *** | −0.385 ** | −0.427 ** | −0.420 ** | −0.045 |
| Active landscape characteristics | Playground | 0.085 | −0.047 | 0.077 | 0.099 | −0.156 |
| | Walking path | 0.194 | 0.143 | 0.247 | 0.219 | 0.272 |
| | Open space | −0.170 | −0.244 * | −0.168 | −0.176 | −0.337 ** |
| | Sports infrastructure | −0.067 | 0.053 | −0.035 | −0.026 | 0.299 ** |
| | Plant viewing | −0.347 | −0.084 | −0.347 | −0.067 | 0.007 |
| | Animal interaction | 0.001 | 0.617 ** | −0.041 | 0.240 | 0.570 * |

* $p < 0.1$, ** $p < 0.05$, and *** $p < 0.01$.

**Table 2.** Results of the *correlation* analysis of the number of sentiment and landscape features.

| Landscape Characteristics | | Sentiment of Tweets | | | |
|---|---|---|---|---|---|
| | | Chinese Sentiment | English Sentiment | Southeast Asian Sentiment | Portuguese Sentiment |
| Physical landscape characteristics | Area | −0.012 | 0.070 * | −0.054 | −0.042 |
| | Plants density | 0.011 | −0.032 | 0.010 | −0.013 |
| | Architecture | 0.038 | 0.097 * | −0.068 | −0.074 |
| | Water | −0.043 | −0.002 | 0.006 | −0.087 |
| Active landscape characteristics | Playground | −0.028 | −0.029 | −0.016 | 0.019 |
| | Walking path | −0.028 | −0.121 ** | 0.024 | −0.122 |
| | Open space | −0.068 ** | −0.097 | 0.007 | −0.092 |
| | Sports infrastructure | 0.059 * | 0.099* | 0.083 | 0.023 |
| | Plant viewing | −0.021 | 0.197 ** | −0.200 ** | −0.161 * |
| | Animal interaction | −0.044 | −0.213 ** | −0.486 *** | −0.089 |

$* p < 0.1$, $** p < 0.05$, and $*** p < 0.01$.

In particular, Portuguese tweets showed a significant positive correlation with plant density and sports infrastructure. Both Southeast-Asian- and Portuguese-language tweets demonstrated a significant negative correlation with open spaces. Regarding activity landscape features, both Chinese and Portuguese tweets exhibited a positive relationship with animal interactions, likely reflecting the deep cultural integration of Chinese and Portuguese influences in Macau's urban landscape preferences.

Regarding sentiment intensity, Chinese tweets showed a negative correlation with open spaces and a positive one with sports infrastructure. English-language tweets displayed a significant positive correlation with park area, architecture, sports infrastructure, and plant landscape, but a negative correlation with walking paths and animal interactions. Southeast-Asian-language tweets had negative correlations with plant landscapes and animal interactions, a pattern also observed in Portuguese-language tweets.

## 5. Discussion

### 5.1. Research Results

This study aims to unravel the emotional impact mechanisms of UGSs in small-scale cities amidst diverse cultural backgrounds. A cross-linguistic sentiment analysis of tweets concerning various types of urban public green spaces revealed the emotional inclinations of different linguistic communities. The results show that in terms of the number of tweets, Chinese-, English-, Portuguese-, and Southeast-Asian-language speakers are not keen on posting water-related tweets; Chinese speakers were more inclined to post tweets related to animal interactions, while Portuguese tweets were positively correlated with posts about plant density and sports infrastructure. In terms of emotional intensity, Chinese tweets on average showed more positive sentiment toward sports infrastructure, while English tweets demonstrated a greater emphasis on park area, architecture, sports infrastructure, and plant landscapes.

Therefore, it is evident that not all green spaces necessarily elicit positive emotions in users. Certain landscape features can inspire positive emotions in visitors to green spaces, while for some groups, the same features may provoke negative reactions, highlighting the diversity in green space perception among different communities.

Overall, examining the emotional positivity of different language speakers toward park landscape features can contribute positively to the study and development of urban

parks in Macau. By better addressing the recreational needs of diverse cultural groups, urban parks can enhance the positive emotional value in high-density urban spaces.

Such findings underscore the importance of abandoning conventional thinking in UGS planning. With cities increasingly becoming melting pots of diverse cultures, leveraging social media data becomes vital for urban planners and local authorities to offer more public engagement options. By fully harnessing public opinions and feedback, we can develop UGS systems that cater to the needs of a multicultural populace. This approach effectively promotes the well-being and positive emotions of cross-cultural residents in urban areas, enhancing the livability of city spaces and contributing to sustainable urban development.

*5.2. Academic Implications*

Firstly, most of the current green-scale research focuses on more typical and widespread cities, yet little focuses on high-density cities. Differences in the distribution and quality of urban green spaces and the availability of green space resources have been widely recognized as an important environmental justice issue, especially in high-density cities [92]. In contrast to this existing research, the same issues of green space equity and environmental justice are present in three ethnically diverse but low-density cities, New York, Phoenix, and Portland, arguing that barriers to accessing green space are related to cultural diversity [93]. Compact urban areas with high population densities are more threatened by energy consumption than low-density cities, which means that fewer people are able to enjoy the benefits of green spaces, and fewer types of green spaces are available. The study of the emotional perception of users of urban green spaces in high-density cities can provide a basis for the evaluation and optimization of urban green spaces at different scales, and provide new ideas for research on public participation.

Secondly, previous studies have analyzed the demand for park use by 300 residents of Turkish culture through a questionnaire [94], and also compared it with other cultural groups, and found significant differences in the focus of different cultural groups on park use, with Turks usually engaging in recreational activities in green spaces as well as having a preference for natural landscapes, whereas in Western countries there is a tendency to prefer interactions with animals or sports activities. Similarly, we have taken into account the different preferences of multicultural populations for the use of urban green spaces, but in contrast to previous studies, this study uses the data from social media platforms as data sources to measure the perception of users' emotions, supported by text-based sentiment analysis methods in machine learning, which compensates for the shortcomings of the traditional methods of obtaining public data by questionnaire surveys and interviews analysis. A method is provided to use the sentiment analysis technique as a tool to quantify the difficult-to-measure emotions for assessing different people's perception of urban green space, and to correspond different cultural background groups with their preferred urban green space landscape features. The visualization is used to explore the interactions between cross-cultural human needs and the provision of different urban green spaces, thus providing guidance and reference for future park construction.

Finally, the comments people post on social platforms are often expressions of real emotions in the moment. The application of automated text analytics to measure themes, ideologies, emotions, and even personalities in fields such as political science and political psychology is booming [95], and in the face of problems such as marginalized groups whose views are ignored, urban space users who have no place to provide claims, or are afraid to make claims, the use of social media allows for the sharing and discussion of views without any restriction [95], and there have been studies that have dealt with the participation of different social cultural groups in the management of urban green spaces, with a focus on immigrant groups with language barriers [96]. In this study, we used sentiment analysis techniques to obtain emotional perceptions of different types of urban green space through undifferentiated expression in social media, invariably giving equal voice to the "respondents", thus making the results of the study more truthful, fair,

and reliable, and providing a positive contribution to the management of urban green space [97].

*5.3. Practical Implications*

This research investigates the differing perceptions of urban park usage among people from various cultural backgrounds, offering a new perspective for analyzing the intricate behaviors associated with multiculturalism in a global society. There are three main contributions:

First, from the perspective of the theory of human–environment fit (PE fit), the pursuit of balance in the field of life is important for the fit between people and the environment in the Chinese context; traditional Chinese culture emphasizes the unity of heaven and man, and the harmonious coexistence of man and nature, and is more inclined to seek integration and balance with the natural environment, especially for Macao, where the phenomenon of aging is more serious, natural environments that allow for outdoor exercise have become the Chinese people's more preferred type of landscape [98]. In the Western context, cultural differences can limit the fit between people and the environment, where there is more emphasis on individualism in Western culture, which tends to emphasize the convenience of activities, practicality, and aesthetics of man-made design in landscape preferences. Therefore will may pay more attention to larger areas of green spaces, buildings, artificial landscapes and planted landscapes [99]. The frequency of use and perceived benefits of urban green space, as an indispensable part of human life, is the key to the search for the fit between people and the environment in the context of Eastern and Western cultures, and can provide a useful research perspective on the harmonious development of Macao [100].

Secondly, not everyone will make the same judgment about the event, which requires the culturally relevant dimensions to be considered [101]. Firstly, with the diversified development of society, because of the differences in the understanding of the environment and nature among people of different cultural backgrounds, it is very important to respect the needs of different cultures for landscape features and to provide diversified green space experiences for the harmonious development of the society, which can be achieved by appropriately increasing the sports infrastructure and enriching the plant design to satisfy the aesthetics of different cultural groups, so that everyone can find a unique natural perception and identity, which can effectively promote social cohesion and improve social well-being [102]. Secondly, the provision of diverse types of green space can become a shared public area, providing opportunities for people of different cultures to communicate harmoniously. It helps to break down barriers between cultures, promotes mutual understanding and respect between people of different races, reduces tense social relationships due to cultural differences, and helps to promote inclusiveness and communion in urban communities [103]. Finally, we found that by studying cities with similar backgrounds, there are examples of studies that show that in-depth research on the perceived role of different types of people in green spaces is very beneficial for the sustainable development of a diverse society. Singapore, as a typical high-density city, believes that green space has an important role in urban development, there is the use of people's health and sustainability, which includes an analysis of the recreational behavior patterns of three different types of racial groups in the same park, as well as tourists, laborers and other groups of people, have agreed on the role of green space in urban development [104]. There is also Hong Kong by investigating the perceived benefits of different types of people on urban greening and studying how to attract global talent and boost tourism by improving the construction and development of urban green space [105]. Therefore, adding the preferences of different groups of people for green space characteristics and studying the diversity of urban green space in depth can produce a special and rich green space development model [106], which attracts residents and tourists of various cultural backgrounds, making it a social gathering place and a center for cultural exchange, and in the long run, it can promote positive group relations and create favorable conditions for social harmony.

Third, worldwide, urban designers and planners need to explore the nature of design through the evaluations of different people, and try to find effective ways to improve the value of design from different evaluations so that design can better serve people. Based on this, the preference for urban green space reflects the process of interaction between the physical characteristics of the landscape and the psychological response of landscape viewers [107]. The results of the study on the differences in the perception of urban green spaces as reflected by people from different cultural backgrounds will help designers and urban green space managers to effectively incorporate public perceptions into their design and decision-making processes. In addition, it provides a perspective that can be referred to for complex landscape perceptions, and the landscape features mentioned in this paper can be studied during the planning process for insight into entry points for future green space planning.

*5.4. Limitations and Future Research*

Additionally, given the myriad of cultural types, future research should extend to an examination of the emotional requisites of various cultural demographics and, more ambitiously, the needs across cultures, which would aid in discerning the interplay between culture and emotional perception, as well as promoting fairness in the utilization of green spaces. Furthermore, linguistic differentiation could be more nuanced; this study did not distinguish between Mandarin and Cantonese due to the sample volume, but subsequent research could incorporate a broader array of regions to dissect these distinctions. This work is merely the inception of cross-cultural perceptual research, offering a broad comparative analysis. Future endeavors could employ more extensive methods to delve into the detailed needs and perceptions characteristic of each cultural and linguistic group.

**6. Conclusions**

Comprehending the perceptual preferences of residents toward urban greenery is vital for urban planning and administration. When investigating the emotional tendencies of different language-speaking populations, analyzing social media data helps provide insight into the cross-cultural public sentiments regarding the use of UGSs. By categorizing social media data linguistically, analyzing sentiments, and visually representing these analyses, we provide a clearer emotional tendency map for various cultural demographics. Moreover, this research analyzed the connection between social media data and the landscape characteristics of UGSs, offering further explanations for the factors that affect the emotional tendencies toward Macau's urban greenery. This provides a foundation for meeting the needs of a culturally diverse populace in their engagement with Macau's green spaces, enhancing the city's social equity and stability, and ensuring the maximization of public interest.

**Author Contributions:** Conceptualization, M.W. and Y.Y.; Methodology, M.W.; Software, M.W.; Validation, Y.Y. and M.L.; Formal analysis, M.W.; Data curation, M.W., Y.Y. and M.L.; Writing—original draft, M.W.; Writing—review & editing, Y.Y. and M.L.; Supervision, Y.Y.; Project administration, Y.Y.; Funding acquisition, L.Z. All authors have read and agreed to the published version of the manuscript.

**Funding:** The authors gratefully acknowledge the support from Macau Science and Technology Development Funds (0067/2022/A).

**Data Availability Statement:** Data is contained within the article.

**Conflicts of Interest:** The authors declare that they have no known competing financial interests or personal relationships that could have appeared to influence the work reported in this paper.

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
