# Peer review of "Differences in Emotional Preferences toward Urban Green Spaces among Various Cultural Groups in Macau and Their Influencing Factors"

_land, doi:10.3390/land13040414_

Round 1

Reviewer 1 Report

Comments and Suggestions for Authors

Review of ISSN 2073-445X

This manuscript characterizes the emotional preferences of different cultures for different types of urban greenspace and landscape elements (e.g., water features, plants, etc). It makes good use of social media data and sentiment analysis to evaluate different cultural preferences and was a very interesting read. The discussion section is the only part of this manuscript that I feel needs major revisions. I was left wondering what the implications of the author’s findings were for urban landscape design (what have we learned about how to design urban spaces better?). I also would like to see a more in-depth discussion of how what they have found compares to other studies that have evaluated cultural landscape preferences. The rest of my comments are more minor (detail level) and have been provided below for each section of the manuscript. My overall recommendation is that this manuscript be considered for publication following major revisions.

Abstract

-       Well written and very clear!

-       It looks like the first word “This” has been put in bold accidentally.

-       I don’t know that you need to justify your use of tools in the abstract – I’d delete “Facing the challenges of timeliness and data acquisition inherent in traditional survey methods” and begin with “We utilized…”

-       I’d also cut “aims to explore” from the first sentence of the abstract and simply say “This study explores”. Aims to makes it sound like you might not actually have done it…

Introduction

-       Line 33: there appears to be a plurality issue (either change urban green space to urban green spaces or change play to plays)

-       Line 43: delete cultural in front of roles – “different roles and symbolic meanings in different cultures” should suffice

-       Research question 2 contains a lot of research question 1. Perhaps rephrase as: (2) How do the emotional reactions people have to UGSs differ among groups from different cultural backgrounds?

Research overview

-       Very thorough. I enjoyed reading this.

-       Line 88: I would reword this a bit. A fair bit of research has been done linking sociodemographic characteristics to perception of UGS in urban planning and landscape architecture. You probably want to cite some of Joan Nassauer’s classic work here. You might also look at https://doi.org/10.1002/pan3.10300 and https://doi.org/10.1002/pan3.10067 for some recent examples of work that has been done in this area. As a side note, the reference you have for this statement in the reference section (33) looks like there is some kind of problem with it (journal details are not provided).

-       Line 97: this sentence does not make sense to me as written. Perhaps delete everything between the two commas and simply say “Urban planning faces the challenge of designing for a culturally heterogenous society”

-       Line 138: I’m not sure what closely interrelated yet distinct terms is referring to. I’m only seeing the one term (emotions). If you are referring to the plurality and complexity of emotions themselves, I’d suggest rewording to make that clear.

-       I was left wondering how sentiment analysis conducted on posted content handles the typical social media challenges (i.e., how does it identify/discard information that is posted by bots, separate false likes/dislikes from comments actually made by real people or interpret sarcastic comments like “this is super! ☹”?). This is obviously not the portion of the paper to go into great detail about these issues, but a sentence or two acknowledging these challenges and how they are typically addressed would be helpful. I see that section 5.4 has a lot of what I was looking for….perhaps move some of the material from 5.4 up to this section…

Methods

-       Line 174: What are the spatial distribution maps of and how will they be used. Please add 1 sentence or so here to clarify this point.

-       In figure 3 it would be helpful if you could indicate how many of each UGS type were present. Just put the number in parentheses after the legend label. For instance: Natural ecological zone (1)

-       Line 241: Can you describe a bit more here how the toolkit library is assessing emotional tone and awarding a sentiment score. It would be helpful to know what goes into this process. It would also be helpful to know what the possible range of the sentiment scores is (its hard to know how large -0.05 – 0.05 really is…)

-       Line 245: Why was the google cloud natural language API used for all languages other than English? Did you try using it for English as well and compare it to the results from the natural language toolkit to ensure that the two approaches were comparable? This seems like an important methodological check

-       Lines 253-260: this is a repeat of the information you provided in the section above. I do not think you need to state it again here

-       Line 277: I’m not sure that I would classify human health as a cultural ecosystem service…

-       Paragraph beginning line 276: This paragraph reads more like background material that does not really belong in the methods section itself. The same goes for the next two sections. I’d work this material into its own section within the research overview. I think it would seem more logical there.

-       Line 287: please be specific – what kinds of health benefits…mental? Physical?

-       Line 305: please describe the 1-7 grades. Do you mean that you created a 1-7 Likert scale with 7 being high quality and 1 being low quality?

-       Figure 9 needs to be condensed so that it does not span 3 pages. I find it difficult to tell the difference between the color gradients in this figure. It would be better to use a viridis or jet color scale so that the differences are more apparent. Color differences across panels matter less than color differences within the panel…

Results

-       Line 341: Did you run a control where you looked at the sentiment of tweets in a highly urbanized area? Is it possible that sentiments are just generally positive…i.e., how do we know that greenspace if responsible for the positive sentiment?

-       Line 353: Again, I find this really interesting, but am curious if this is something that is generally true (are Portuguese tweets always more positive or are they just more positive in greenspace….similarly are English tweets always less positive or just less positive in greenspace?

-       Its not possible to read the x-axis on Figs 19 and 20.

-       Fig 19-22: it would be helpful to use the same color scheme each time you refer to a particular language. It would be helpful to refer to your figures in the text that you right (i.e., call out Figs 19-22 when you describe a pattern that can be visualized in one of them)

-       Line 349: I’m not seeing that Portuguese tweets have more positive sentiment for natural ecological zones. Given Fig. 22 it actually looks like Portuguese tweets have less positive sentiment for natural ecological zone than any other UGS. Same thing for some of the other claims in this paragraph (they do not appear to be consistent with Fig. 22)

-       Line 395: please explain how a positive correlation with animal interactions is reflective of the influence of Chinese and Portuguese culture on Macau’s urban landscape preferences…why animal interactions in particular?

-       Lines 405-408 don’t belong in the results section because they are not a result.

Discussion

-       Lines 421-422. This sentence reads as if it is saying that green spaces with water were associated with negative sentiments, but that’s not the case in your table. They did receive fewer tweets which is not the same thing as tweets expressing negative sentiments. I think it is important to clarify your language here.

-       Lines 442-445 are a repeat of what is written in line 413. Please delete the repeat

-       The paragraph beginning on line 446 is also repetitive, restating material from section 5.1. Please delete repeated text

-       The material provided in section 5.2 does not really indicate why what has been found is meaningful. I would like to see the research results tied back to other work that has looked at UGS perspectives of different cultures. Work of this sort has been done previously even if its not prevalent. How is what you have found different? How is it similar? Right now section 5.2 reads more like a pitch for why the study was done in the first place than what it implies, which is not really what you want out of a discussion section.

-       Lines 474-479: this is interesting, but please indicate how it is supported by the specifics of what you found. Why is striving for balance important within Chinese settings (how is this indicated by their landscape preferences). How is the opposite indicated by the landscape preferences of western cultures. Please be specific here.

-       Section 5.3: its not clear to me how what is being discussed here ties back to your results. How does your finding that the sentiment of tweets for plant viewing is significant and positive for English speaking cultures but negative for southeast Asian and Portuguese cultures inspire a discussion of mitigating racial tensions and advancing our comprehension of the human essence? If your aim is to say that because different cultures value different aspects of landscapes and the societies we live in are increasingly multicultural, it is important to offer diverse greenspace experiences to promote societal wellbeing and reduce cross-cultural tension, say so directly. And if this is the point you want to make, spend some time unpacking this claim. Stating that diverse landscapes might reduce cross-cultural tension is one thing, digging into the literature and finding examples of how this might work in practice is another.

-       I’m left wondering what you think the implications of your findings are for a UGS design perspective. What should we be promoting as urban planners?

Comments on the Quality of English Language

I have included minor English language suggestions in my "comments/suggestions for the authors" file (see above) when I have detected plurality issues or areas of the document with repetitive text.

Author Response

We greatly appreciate your thorough and comprehensive evaluation of our articles. Your insights have highlighted key areas for our improvement, and we are committed to advancing in these aspects. Please see the specific revisions in the attachment.

Reviewer 2 Report

Comments and Suggestions for Authors

In light of the growing trend of cultural integration, this study intends to investigate the variety of emotional inclinations and demands towards urban green areas among individuals from various cultural backgrounds.

The authors have analyzed geolocation data and social media comments from twitter from Macau region. Regression analysis and Correlation analysis has been performed, correlating  ten locally relevant landscape features with social media comments.

The article is very interesting. The level of English is very good, all data has been properly reported. Outcomes are based on statistical analysis. The article follows the typical sequence of academic writing. The research methodology scheme is very comprehensive.

This research’s motivation and novelty are clearly stated and the findings and the practical implications have been clearly stated. Limitations and future work are included at the end of this manuscript

Although this article is very well written, some comments are addressed to the authors before publication

1.      It would be usefull for the readers to fond the research questions in form of RQ1, RQ2… at the end of the introduction section. Consequently, the findings of those RQ should be directly responded in the conclusion section

2.      The similarity report in Turnitin  has shown 29% plagiarism. Nevertheless, most of the highlighted text in the report represents terms that may not be paraphrased.

3.      The literature review is quite rich. The 91 articles cited are overall related to the manuscript’s content. 23 out of the 91 citations have been published more than one decade ago. The authors are advised to review those 23 cited articles and examine the possibility of replacing those citations with newer ones

The authors are also advised to study the following articles

Sinou, M.; Skalkou, K.; Perakaki, R.; Jacques, S.; Kanetaki, Z. Holistic Strategies Based on Heritage, Environmental, Sensory Analysis and Mapping for Sustainable Coastal Design. Sustainability 202315, 9953. https://doi.org/10.3390/su15139953

Droumeva, M. (2017). Soundmapping as critical cartography: Engaging publics in listening to the environment. Communication and the Public2(4), 335-351. https://doi.org/10.1177/2057047317719469

Overall, this article is very good, the topic very interesting and fits in the scope of the Special Issue.

Author Response

(The authors gave the same response as above.)

Reviewer 3 Report

Comments and Suggestions for Authors

Urban green spaces play a very important role in the development of the city, as they perform recreational functions. Thousands of citizens visit parks every day, where they want to spend time with friends and family members, breathe fresh air, restore their strength after work or on weekends. Visiting such places evokes certain emotions among city residents, and it is very important for local authorities that these emotions are positive.

This article is devoted to the analysis of the emotional preferences of residents of different cultural groups in Macau (China) to urban green spaces. The authors conducted a thorough analysis of previous studies, characterized the city of Macau, typified the city’s green areas, processed comments in social networks to understand the emotional perception of green urban spaces by different cultural groups (analysis of comments in different languages) using methods of social network analysis, correlation-regression analysis, programming and GIS technologies.

The article was performed at a sufficient theoretical and methodological level, offers an original approach to the study of urban green spaces and believe that the article worth to be published, but there are several questions for the authors:

• It is not clear what the graph demonstates in fig. 19, 20. What does "Greenspace Code" mean?

• It is not entirely clear what the physical and active landscape characteristics reflect, and with which the regression analysis was conducted (area, plan density, architecture, etc.) (tables 1, 2).

• Judging by the coefficients presented in tables 1,2, we are talking about correlation coefficients. Therefore, apparently, the table characterizes the results of correlation analysis, not regression one.

Author Response

(The authors gave the same response as above.)
